# SEAL: Self-supervised Embodied Active Learning using Exploration and 3D Consistency

**Devendra Singh Chaplot**[1]*, **Murtaza Dalal**[2], **Saurabh Gupta**[3],
**Jitendra Malik**[1,4], **Ruslan Salakhutdinov**[2],
[1]Facebook AI Research, [2]Carnegie Mellon University, [3]UIUC, [4]UC Berkeley

Project Webpage: https://devendrachaplot.github.io/projects/seal

## Abstract

In this paper, we explore how we can build upon the data and models of Internet images and use them to adapt to robot vision without requiring any extra labels. We present a framework called Self-supervised Embodied Active Learning (SEAL). It utilizes perception models trained on internet images to learn an active exploration policy. The observations gathered by this exploration policy are labelled using 3D consistency and used to improve the perception model. We build and utilize 3D semantic maps to learn both action and perception in a completely self-supervised manner. The semantic map is used to compute an intrinsic motivation reward for training the exploration policy and for labelling the agent observations using spatio-temporal 3D consistency and label propagation. We demonstrate that the SEAL framework can be used to close the action-perception loop: it improves object detection and instance segmentation performance of a pretrained perception model by just moving around in training environments and the improved perception model can be used to improve Object Goal Navigation.

## 1   Introduction

Even though computer vision started out as a field to aid embodied agents (robots) [21], in current times it has evolved into *Internet computer vision*, where the focus is on training models for and from Internet data. Fueled by the successes of supervised learning, today's models can successfully classify, detect and segment out objects in Internet images reasonably well [19]. This raises a natural question: would we need to restart from scratch and gather similarly large-scale labeled datasets to get computer vision to work for embodied agents, or can we somehow bootstrap off the progress made in Internet computer vision?

Internet data comprises of sparse and unrelated snapshots of the world, carefully chosen by humans. On the one hand, this simplifies the problem as models only need to reason about a small and well-chosen subset of possible views of the world. On the other, this makes learning hard. The dog in `image_0032` in the ImageNet dataset, is different from the dog in `image_0033`, and there is no way to understand how the dog in `image_0032` will look like from another view. In contrast, an active agent, embodied in a 3D environment, experiences views of a 3D consistent world. Spatio-temporal continuity in views of the underlying 3D world can allow better inference at *test* time, and label efficient learning at *train* time. These effects are further amplified, as the agent can actively choose the views it experiences to maximize learning. Thus, in contrast to Internet computer vision, embodiment opens up the possibility of deriving supervision from spatio-temporal continuity and interaction.

Such self-supervision for physical tasks (*e.g.* collisions, graspability) can be done through the use of appropriate sensors [17, 36]. How to do so for semantic tasks, such as segmenting and detecting objects is less clear, and is the focus of our work in this paper. We build off models from Internet computer vision and show how their performance can be improved through self-supervised interaction.

---

*Correspondence: dchaplot@fb.com

35th Conference on Neural Information Processing Systems (NeurIPS 2021).

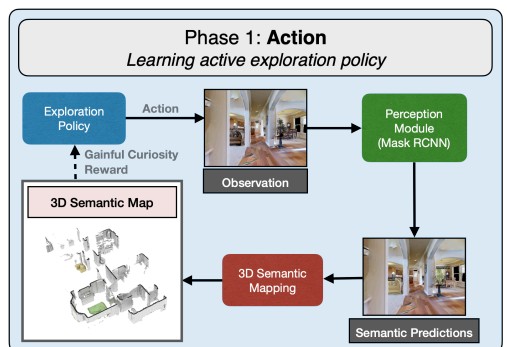 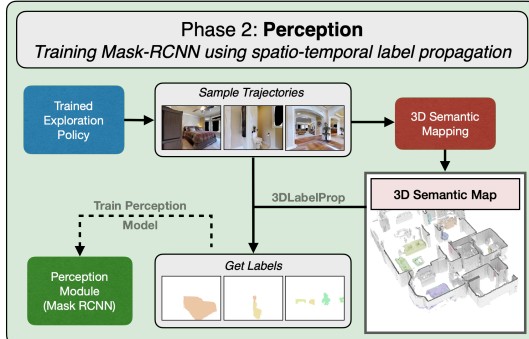

**Figure 1: Self-supervised Embodied Active Learning.** Our framework called Self-supervised Embodied Active Learning (SEAL) consists of two phases, *Action*, where we learn an active exploration policy, and *Perception*, where we train the Perception Model on data gathered using the exploration policy and labels obtained using spatio-temporal label propagation. Both action and perception are learnt in a completely self-supervised manner without requiring access to the ground-truth semantic annotations or map information.

We show that a) this can be done purely via interaction, *without needing any additional supervision*, b) improved models exhibit better performance in *test* environments *as is*, c) models can be improved further through small amounts of self-supervised interaction in the *test* environment. We also show that improved perception can, in turn, improve the performance at interactive tasks.

We propose a framework called Self-supervised Embodied Active Learning (SEAL) as shown in Figure 1. It consists of two phases, one for learning Action, and another for learning Perception. During the *Action* phase, the agent learns a self-supervised exploration policy to gather observations of objects with at least one highly confident viewpoint. We propose an intrinsic motivation reward called Gainful Curiosity to train this policy. In the *Perception* phase, the learned exploration policy is used to gather a single episode of observations in an environment. We propose a method called 3DLabelProp to obtain labels for these observations in a self-supervised fashion. Both the action and perception phases involve constructing a 3D Semantic Map. The agent observations in an episode are used to construct an episodic 3D semantic map. The map is used to compute the Gainful Curiosity reward during the Action phase. And during the perception phase, the semantic labels in the 3D map are projected on the agent's original observations using 3DLabelProp to generate the supervision for training the embodied perception model.

Our experiments demonstrate that the SEAL framework can be used to close the action-perception loop. Perceptual models allow the agent to act in the world and collect data that improve the perception models. Improved perception models can, in turn, improve the agent's policy for interacting with the world. We first use SEAL framework to improve object detection and instance segmentation performance of a pretrained perception model (Mask RCNN [19]) from 34.82/32.54 AP50 scores to 40.02/36.23 AP50 scores by just moving around in training environments, without having access to any additional human annotations. By allowing the agent to explore the test environment for a single episode, we can further improve the performance to 41.23/37.28 AP50 scores. Next, we also show that this improved perception model can be used to improve the performance of an embodied agent at Object Goal Navigation from 54.4% to 62.7% success rate.

## 2 Related Work

In this paper, we propose a technique for self-supervised improvement of perception through active interaction, exploration, and 3D semantic mapping. Many papers tackle related problems and we survey them here.

**Self-supervised Learning.** A number of papers focus on the design of label-free pretext tasks to pre-train visual representations [1, 24, 57]. Most such works focus on learning a good generic feature representation without access to any semantic labels. Our work is also self-supervised, we also don't rely on any external source of supervision during training. However, instead of producing a generic feature representation, we produce improved object detection and segmentation models through self-supervision. Our supervision comes from the 3D self-consistency of a given perception model on different views of the data. Furthermore, we learn to actively seek data to conduct this self-supervision on *vs.* past work that employs pre-collected datasets.

**Domain Adaptation.** Another related line of work is that of unsupervised domain adaptation [20, 47, 39], where the focus is to adapt models trained on one domain to work well on another domain without any labels, or any assumptions on the data that is available in the two domains. Our focus here is orthogonal: we specifically focus on opportunities for self-supervision available in the context of embodied agents and learn policies to gather data in the target domain that eases transfer. General domain adaptation techniques could be used on top of our work.

**Active Perception and Learning.** Active interaction with an environment can improve performance at *test* time by gaining information from better views (*i.e. active perception* [6, 2]), as well as at *train* time to generate more training data or to mine harder examples for external labeling (*i.e. active learning* [44]). Active learning and active perception have largely been thought of as two separate bodies of research. Our proposed approach tackles both these problems, and we highlight key differences from existing research.

Work in active learning focuses on the selection of data points from an unlabeled corpus for labeling by an oracle (*e.g.* humans). Researchers have explored the use of prediction uncertainty [45, 16], coverage [42] and meta-learning [28, 56] to learn such data selection functions. Our work departs from standard active learning in two ways: we do not assume access to a pre-collected corpus of unlabeled data, nor to a labeling oracle. We learn a policy to efficiently and autonomously acquire unlabeled data, and rely on multi-view 3D consistency to derive supervision. This differentiates our work from work in active learning [37, 44], and also recent works that relax one or the other requirements: the need for oracles using spatio-temporal consistency to improve models using pre-collected unlabeled videos [38, 46, 27], or oracle supervised active learning in embodied environments [11, 33].

Similarly, active perception has been studied over the last many years [6, 2]. Early methods used information-theoretic measures to determine the next best view [12]. More recently focus has been on learning a policy in an end-to-end fashion to directly improve metrics of interest [25, 55, 26, 3, 54]. There also has been some work to improve segmentation and object detection using Simultaneous Localization and Mapping (SLAM) [58, 48]. The above works learn active interaction with an environment to improve performance at test time by gaining information from better views. In contrast, our focus is on active learning through interaction, i.e. learning how to explore an environment at train time to automatically generate more training data which is used to improve perception. Our proposed technique can also be used to scale up active perception to effectively detect and segment objects in a large-scale 3D scene at test time (see the specialization setting in Sec 3). Rather than exploring each object one at a time, we learn to explore the whole 3D scene at once. This improves recognition performance for all objects in the scene in a single episode. Furthermore, unlike many prior methods, our approach is completely self-supervised.

**Learning through Interaction.** Robot learning and reinforcement learning focus on learning to solve interactive tasks directly through hit and trial interaction. However, there is also a small body of work that seeks to improve perception models through unsupervised or self-supervised interaction. Unlike prior work [13, 35, 23] which tackle bottom-up image segmentation or learning object representations in table-top manipulation setting, we learn to improve object detection and instance segmentation models in the context a mobile navigation agent. Parallel work from Fang et al. [14] also focuses on improving object detection models using active data. Unlike their work, we learn a policy for data collection, and aggregate information using 3D semantic maps.

**3D Semantic Mapping.** 3D mapping (reconstruction and localization) has been well studied, and is a fairly mature sub-field of robotics and computer vision. We refer the readers to Fuentes-Pacheco et al. [15] for a survey. Researchers have also considered the task of associating semantics with 3D maps [32, 8]. We adopt and adapt these ideas to our setting, and focus on how we could use these representations for learning active exploration policies to improve models for embodied perception.

## 3 Method

Our objective is to train an embodied agent to learn both action and perception by moving around in a physical environment. We assume the agent is given access to a perception (object detection and instance segmentation) model, $f_P$, such as a MaskRCNN [19] pretrained on static Internet data. The agent needs to learn a policy to move in the environment and use the experience to learn embodied perception in a completely self-supervised manner without having access to the ground-truth semantic annotations or map information.

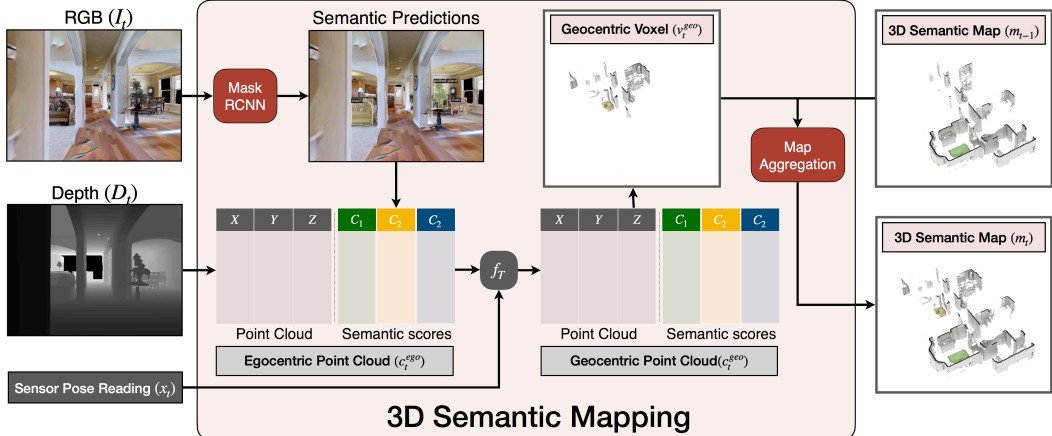

**Figure 2: 3D Semantic Mapping.** The 3D Semantic Mapping module takes in a sequence of RGB ($I_t$) and Depth ($D_t$) images and produces a 3D Semantic Map.

We propose a framework called **Self-supervised Embodied Active Learning (SEAL)** to tackle this problem. As shown in Figure 1, it consists of two phases, one for learning Action, and the other for learning Perception. During the *Action* phase, the agent learns a self-supervised exploration policy to gather useful observations. In the *Perception* phase, the learned exploration policy is used to gather a single episode of observations in an environment. The observations are labeled in self-supervised manner using 3D Label Propagation, which is then used for training the embodied perception model. In our framework, both the Action and Perception phases require building a 3D semantic map from a sequence of agent observations. We first describe the 3D Semantic Mapping module (Sec 3.1) and then describe how it is utilized in both the phases to train the active exploration policy (Sec 3.2) and the embodied perception model (Sec 3.3).

**Generalization and Specialization.** We employ this framework under two settings, Generalization and Specialization. In the *Generalization* setting, the agent is allowed to train for 10 million frames in the set of training environments and tested directly on a set of unseen test environments. In the *Specialization* setting, after training, the agent is also allowed to explore each test environment for a single episode of $T(=300)$ time steps. In both settings, the agent has to learn completely in a self-supervised manner, without having access to the ground-truth semantic annotations or maps in training or test environments. For the Generalization setting, the trained perception model is tested on random images in the unseen test environments. For the Specialization setting, the second phase is repeated for each test environment before testing on the unseen images in the test environment.

**Agent Specification.** We follow the agent specification from Chaplot et al. [9]. For each time step $t$, the observation space consists of an RGB observation, $I_t \in \mathbb{R}^{3 \times W_I \times H_I}$, a depth observation, $D_t \in \mathbb{R}^{3 \times W_I \times H_I}$ and a 3-DOF pose sensor $x_t \in \mathbb{R}^3$ denoting the $x$ and $y$ coordinates of the agent and the orientation of the agent. The action space consists of 3 discrete actions: move forward ($25cm$), turn left ($30°$) and turn right ($30°$). The agent camera height is $88cm$.

## 3.1   3D Semantic Mapping

We use a voxel-based representation for the semantic 3D Map. The semantic map, $m$, is a 4D tensor of size $K \times L \times W \times H$, where $L, W, H$, denote the 3 spatial dimensions, and $K = C + 1$, where $C$ is the number of semantic object categories. Among the $K$ channels, the first channel denotes whether the corresponding voxel (x-y-z location) is occupied or not and each of $C$ channels stores the score (between 0 and 1) of the corresponding voxel belonging to a particular object category. We use a cubic voxel of size $(5cm)^3$.

Figure 2 shows an overview of the 3D Semantic Mapping module. The map is initialized with all zeros at the beginning of an episode, $m_0 = [0]^{K \times L \times W \times H}$. The agent always starts at the center of the map facing east at the beginning of the episode, $x_0 = (L/2, W/2, 0.0)$. At each time step $t$, the pretrained MaskRCNN [19] ($f_P$) is to used to get semantic predictions from the input RGB observation, $I_t$. The semantic predictions consist of the score for each pixel belonging to a particular category obtained directly from the Mask-RCNN. If there are multiple predictions for the same pixel, we take the maximum score for each category across all the predictions. The depth observation,

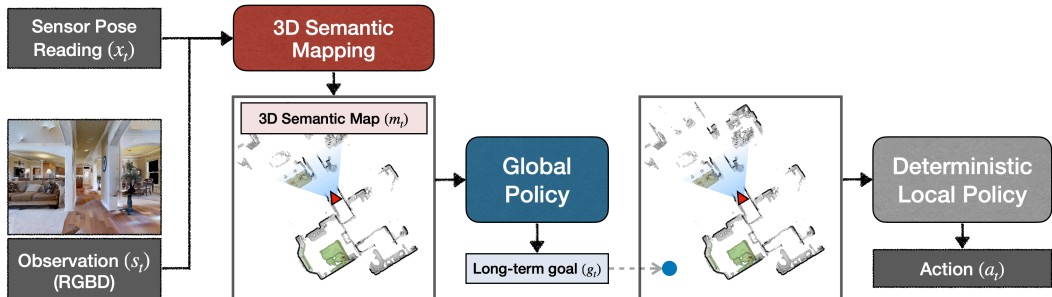

**Figure 3: Learning Action using Gainful Curiosity.** We use a modular architecture for the Gainful Curiosity Policy. The 3D Semantic Mapping module is used to construct and update the map, $m_t$, at each time step $t$. The Global Policy is used to sample long-term goal ($g_t$). A deterministic Local Policy is used to plan a path to the long-term goal and take low-level navigational actions. The Gainful Curiosity intrinsic motivation reward is computed using the 3D Semantic Map.

$D_t$, is used to compute an egocentric point cloud, $c_t^{ego}$. Each point in this egocentric point cloud is associated with the corresponding semantic predictions. The egocentric point cloud is converted to a geocentric point cloud, $c_t^{geo}$ using geometric transformations based on the agent pose, $x_t$, which is then converted to a geocentric voxel representation $v_t^{geo} \in \mathbb{R}^{K \times L \times W \times H}$ using geometric projections. In the voxel representation, the first channel among $K$ channels represents occupancy, and the rest of the $C$ channels denote the maximum score for the voxel belonging to the corresponding semantic category. This voxel representation is aggregated over time using channel-wise max pooling to get the 3D Semantic Map.

## 3.2   Learning Action

The active exploration policy in SEAL needs to explore the environment to gather useful observations for learning perception. Intuitively, we would like the agent to explore as many objects as possible with a highly confident prediction for each object from at least one viewpoint. We require at least one view with high confidence as the agent needs to learn in a self-supervised fashion. In the next subsection, we describe how a highly confident prediction from one viewpoint can be used to label other viewpoints using 3D consistency.

**Gainful curiosity.** We define an intrinsic motivation reward called Gainful Curiosity to train the active exploration policy to learn such behavior of maximizing exploration of objects with high confidence. We define $\hat{s}(= 0.9)$ to be the score threshold for confident predictions. The Gainful Curiosity reward is then defined to be the number of voxels in the 3D Semantic Map having greater than $\hat{s}$ score for at least one semantic category. This reward encourages the agent to find new objects and keep looking at the object from different viewpoints until it gets a highly confident prediction for the object from at least one viewpoint. It also encourages the agent to not spend time exploring walls and corners as they do not belong to any object category. Unlike prior formulations of intrinsic motivation, such as prediction-error curiosity [34] or temporal inconsistency-based semantic curiosity [11] which aim to maximize uncertainty, Gainful Curiosity aims to gain definitive knowledge.

We use a modular architecture for the Gainful Curiosity policy inspired by prior work on modular learning models for visual navigation [10, 9]. The architecture shown in Figure 3 consists of the 3D Semantic Mapping module to build the map as described earlier. The 3D Semantic Map ($m_t$) is passed as input to a Global Policy which selects a long-term goal ($g_t$) or a waypoint, which is essentially an x-y coordinate of the map. The Global Policy consists of convolutional layers followed by fully connected layers. A deterministic Local Policy, which uses the Fast Marching Method [43] for path planning, is used to navigate to the long-term goal using low-level navigation actions. The Global Policy operates at a coarse time scale, sampling a goal every 25 local steps. It is trained to maximize the intrinsic motivation reward with reinforcement learning.

## 3.3   Learning Perception

The trained exploration policy is used to sample trajectories in the training environments. For each trajectory, we build the 3D Semantic Map as described above. The semantic predictions from the Perception Model ($f_P$) for the observations in this trajectory might contain both false positives (detecting an object when there is no object or predicting the wrong object category) and false

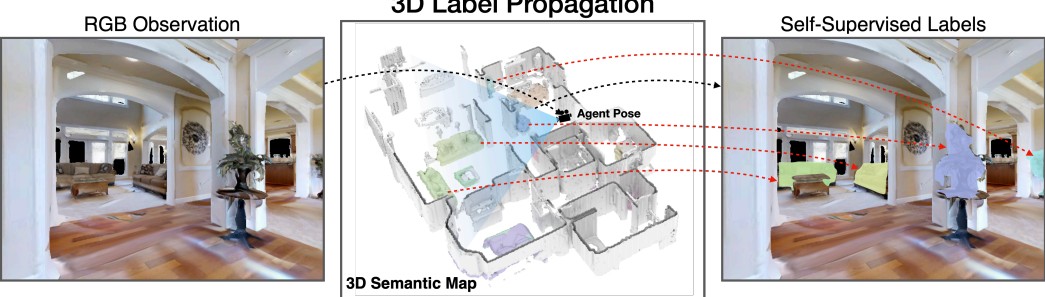

**Figure 4: Learning Perception using 3DLabelProp.** The agent trajectory is to used to create a semantic 3D map of the environment. The map is labelled in a self-supervised manner using 3D consistency. The label for each pixel in the agent trajectory is obtained using ray-tracing in the labeled map based on the agent's pose.

negatives (not detecting an object when there is an object present). Consequently, the 3D Semantic Map will contain ambiguities with scores for multiple object categories for a given voxel that may or may not belong to an object.

**3DLabelProp.** We propose a method called 3DLabelProp to obtain self-supervised labels from the 3D Semantic map. First, to perform disambiguation, we label each voxel to belong to the category with the maximum score above $\hat{s}$. If all categories have a score less than $\hat{s}$ for a voxel, we label it as not belonging to any object category. After labeling each voxel in the map, we find the set of connected voxels labeled with the same category to find object instances. We fill small holes ($< 0.25m^3$) in object instances and remove small objects ($< 0.025m^3$) to get the final labeled 3D semantic map. The instance label for each pixel in each observation in the trajectory is then obtained using ray-tracing in the labeled 3D map based on the agent's pose as shown in Figure 4. Pixel-wise instance labels are used to obtain masks and bounding boxes for each instance. Note that this labeling process is completely self-supervised and does not require any human annotation. The set of observations and self-supervised labels are used to fine-tune the pre-trained perception model.

# 4 Experiments

**Setup.** We use the Habitat simulator [40] with the Gibson dataset [50] for our experiments. The Gibson dataset consists of scenes that are 3D reconstructions of real-world environments. We use a set of 30 scenes from the Gibson tiny set for our experiments whose semantic annotations are available from Armeni et al. [5]. We use a split of 25 and 5 scenes for training and testing identical to prior work [9]. The list of training and test scenes is provided in the supplementary material. Following the setup in prior work [11, 9], we use 6 common indoor object categories for all our experiments: chair, couch, bed, toilet, TV, and potted plant. We randomly sample a set of 2500 images (500 per test scene) and evaluate the final perception model trained using SEAL and the baselines on object detection and instance segmentation tasks. The same test set is used for all the methods and for both Generalization and Specialization settings. We report bounding box and mask AP50 scores for object detection and instance segmentation, respectively. AP50 is the average precision with at least 50% IOU. IOU is defined to be the intersection over union of the predicted and ground-truth bounding box or the segmentation mask.

## 4.1 Hyperparameter and Architecture Details

**Perception Model.** We use a Mask-RCNN [19] using Feature Pyramid Networks [31] with a ResNet-50 [18] backbone as the Perception model. The Mask-RCNN is pretrained on the MS-COCO dataset [30] for object detection and instance segmentation. During the Perception phase, we fine-tune the Mask-RCNN on the data gathered by the Gainful Curiosity policy and labeled using 3D Label Propagation. We use Stochastic Gradient Descent [7] with a fixed learning rate of 0.0001 for $N = 5000$ iterations. All other hyperparameters are set to default settings in Detectron2 [49].

**Action Model.** In the Gainful Curiosity model, the Global Policy is the only learnable component. It is a 5 layer convolutional network ($2\times3D$ convolution layers + $3\times2D$ convolution layers) followed by 3 fully connected layers. In addition to the 3D Semantic map, we also pass the agent orientation as a separate input, which is processed using an Embedding layer and added as an input to the fully-connected layers. The Global Policy is trained using Proximal Policy Optimization [41] algorithm

**Table 1: Results.** Performance of all the baselines as compared to the proposed SEAL framework for both Generatlization and Specialization settings. We report bounding box and mask AP50 scores for Object Detection and Instance Segmentation.

| Method | Generalization | | Specialization | |
|---|---|---|---|---|
| | Object Detection | Instance Segmentation | Object Detection | Instance Segmentation |
| Pretrained Mask-RCNN | 34.82 | 32.54 | 34.82 | 32.54 |
| Random Policy + Self-training [52] | 33.41 | 31.89 | 34.11 | 31.23 |
| Random Policy + Optical Flow [22] | 33.97 | 32.34 | 34.33 | 32.22 |
| Frontier Exploration [53] + Self-training [52] | 33.78 | 32.45 | 33.29 | 32.50 |
| Frontier Exploration [53] + Optical Flow [22] | 35.22 | 31.90 | 34.19 | 32.12 |
| Active Neural SLAM [10] + Self-training [52] | 34.35 | 31.20 | 34.84 | 32.44 |
| Active Neural SLAM [10] + Optical Flow [22] | 35.85 | 32.22 | 35.90 | 33.12 |
| Semantic Curiosity [11] + Self-training [52] | 35.04 | 32.19 | 35.23 | 32.88 |
| Semantic Curiosity [11] + Optical Flow [22] | 35.61 | 32.57 | 35.71 | 33.29 |
| SEAL | **40.02** | **36.23** | **41.23** | **37.28** |

with 25 parallel threads, with each thread using one scene in the training set. We use a time horizon of 20 steps, 12 mini-batches, and 4 epochs in each PPO update. Our PPO implementation is based on [29]. The policy is trained with the Gainful Curiosity reward which is computed by counting the the number of voxels explored with $\hat{s}(= 0.9)$ score for at least one object category. We use Adam optimizer with a learning rate of $0.000025$, a discount factor of $\gamma = 0.99$, an entropy coefficient of $0.001$, value loss coefficient of $0.5$ for training the Global Policy.

## 4.2 Baselines

We are not aware of any methods directly comparable to the proposed method. We adapt prior methods as separate baselines for Action and Perception phases and compare combinations of these baselines to our SEAL framework. We implement the following **Action baselines**:

**- Random Policy.** A policy taking a random navigation action at each time step.

**- Frontier Exploration** [53]. This baseline uses the classical frontier-based exploration heuristic of navigating to the nearest unexplored point to explore unseen environments.

**- Active Neural SLAM** [10]. Similar to our Gainful Curiosity policy, Active Neural SLAM is a also modular map-based learning method. It builds a spatial top-down map and learns a higher-level global policy to select waypoints in the top-down map space to maximize area coverage.

**- Semantic Curiosity** [11]. This baseline is closest to our Gainful Curiosity policy, which is trained to maximize the temporal inconsistency in object detections and segmentation in a trajectory.

We implement the following **Perception baselines**:

**- Self-Training.** In this baseline, we train a Mask-RCNN on its own predictions using the data collected by the exploration policy. This baseline is adapted from prior work on self-training which shows improvement in image classification by training on large datasets of unlabelled images [52, 51].

**- Optical Flow.** In this baseline, we use optical flow [22] between consecutive images for label propagation. Each pixel in the current image is labeled with the maximum score prediction of the pre-trained Mask-RCNN over associated pixels in the previous, current, and next image. Optical flow is also used by Eitel et al. [13] for learning segmentation in an interactive manipulation setting.

## 5 Results

We report the performance of the proposed SEAL framework and all the baselines for both the Generalization and Specialization settings in Table 1. SEAL outperforms all the baselines by a large margin, 40.02/36.23 vs 35.85/32.57 AP50 score for Generalization, and 41.23/37.28 vs 35.90/33.29 AP50 score for Specialization. SEAL also considerably improves over the Pretrained Mask-RCNN baseline (41.23/37.28 vs 34.82/32.54 AP50 score for Specialization) while all the other baselines perform worse or comparable to the Pretrained Mask-RCNN baseline. This indicates that the SEAL Framework can be used to explore a physical environment and improve perception in a completely self-supervised manner without requiring any human annotation.

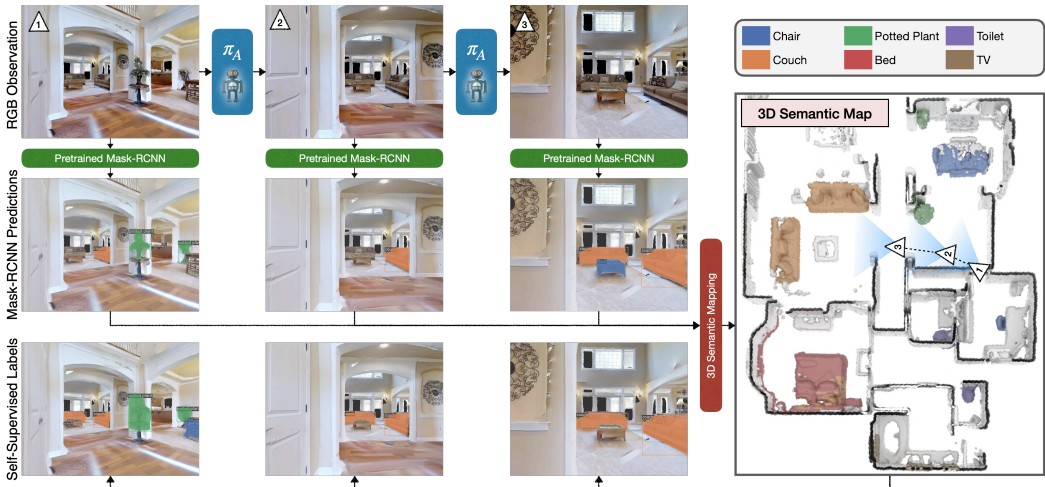

**Figure 5: Example.** Figure showing 3 frames in a trajectory gathered using the trained active exploration policy, $\pi_A$, the corresponding pretrained Mask-RCNN predictions, and the self-supervised labels obtained using the 3D Semantic Map. (1) The Mask-RCNN predictions contain *false negatives*, it fails to detect the couch and the chair, (2) Mask-RCNN still fails to detect the couch, (3) MaskRCNN detects the couch but contains a *false positive* prediction of the coffee table as a chair. The 3D Semantic Map (shown on the right) obtained using the MaskRCNN predictions correctly consists of the couch and the chair. Self-supervised labels obtained using the semantic map correctly identify the objects at all the 3 timesteps.

**Qualitative example.** In Figure 5, we show an example trajectory and corresponding trajectory obtained through SEAL. The pretrained Mask-RCNN predictions (shown in the second row) contain both false positives and false negatives and are inconsistent over time as they are based on individual images. Since the MaskRCNN correctly recognizes the objects with a high score from at least one viewpoint, they are correctly labeled in the 3D semantic map. Consequently, the self-supervised labels obtained from the map using ray-tracing are accurate and consistent over time. More examples are provided in the supplement.

**Ablations.** To understand and quantify the importance of both the Action and Perception phases in SEAL, we perform experiments with several ablations. In each ablation, we replace the Action or the Perception phase in SEAL with a corresponding baseline. Results in Table 4 indicate that both the Action and Perception phases are important for the overall performance. The Perception phase is more critical as replacing it with Self-training or Optical Flow drops the performance comparable to the Pretrained Mask-RCNN. As noted in prior work [52], Self-Training requires large amounts (millions of images) of unlabelled data to be effective which is not available in our setting. The performance of Optical Flow is also limited as it can only aggregate information over consecutive frames. This indicates that aggregating information using semantic 3D mapping and self-supervised labeling using ray-tracing is crucial for the overall performance of the framework.

## 5.1 Closing the Action-Perception Loop: Object Goal Navigation

We demonstrated that the SEAL framework can be used to learn an active exploration policy to gather data and improve perception models, i.e. we used action to improve perception. In order to close the perception-action loop i.e. to use perception to improve action, we deploy the SEAL-improved perception model to a downstream Object Goal Navigation task. We utilize the SemExp model from Chaplot et al. [9] and replace the pretrained Mask-RCNN in SemExp with the SEAL-improved perception model. All the other model components and the experimental

**Table 2: Object Goal Navigation.** Performance of SEAL-improved perception model on the Object Goal Navigation task. SEAL (Gen.) and SEAL (Spec.) refer to SEAL perception models from Generalization and Specialization settings.

| Method | Success | SPL |
|---|---|---|
| SemExp [9] | 0.544 | 0.199 |
| SemExp + SEAL (Gen.) | 0.611 | 0.323 |
| SemExp + SEAL (Spec.) | **0.627** | **0.331** |

setup is identical to [9]. As shown in Table 2, SEAL leads to large improvement, the SemExp + SEAL Specialization perception model led to a success rate of 0.627 and a SPL [4] of 0.331 as compared to 0.544 Success and 0.199 SPL for the SemExp model with the pretrained Mask-RCNN.

**Table 4: Ablation Results.** Performance of all the ablations of the SEAL framework for both Generatlization and Specialization settings. In each ablation, we replace the Action or Perception phase in SEAL with a baseline. We report bounding box and mask AP50 scores for Object Detection and Instance Segmentation.

| Method | Generalization | | Specialization | |
|---|---|---|---|---|
| | Object Detection | Instance Segmentation | Object Detection | Instance Segmentation |
| Pretrained Mask-RCNN | 34.82 | 32.54 | 34.82 | 32.54 |
| SEAL w/o Action + Random Policy | 35.43 | 31.22 | 35.77 | 31.79 |
| SEAL w/o Action + Frontier Exploration [53] | 37.39 | 33.49 | 37.99 | 34.55 |
| SEAL w/o Action + Active Neural SLAM [10] | 38.90 | 34.99 | 39.01 | 35.41 |
| SEAL w/o Action + Semantic Curiosity [11] | 38.39 | 35.20 | 39.21 | 35.62 |
| SEAL w/o Perception + Self-training [52] | 35.35 | 32.47 | 35.88 | 33.20 |
| SEAL w/o Perception + Optical Flow [22] | 35.65 | 32.49 | 35.92 | 33.44 |
| SEAL | **40.02** | **36.23** | **41.23** | **37.28** |

## 5.2 Extension: Weak Supervision

We demonstrated that the SEAL framework can be used to improve perception and action in a self-supervised fashion. Additionally, it can also be used under the weak supervision setting, where few frames in each test environment are annotated by humans. For each annotated frame, we can simply replace the pretrained MaskRCNN predictions in the Perception phase with human annotations with a score of 1. We sample $k$ frames with the highest average entropy over voxels corresponding to all pixels in the 3D semantic map and assume

**Table 3: Weak Supervision Results.** Performance of a Mask-RCNN naively fine-tuned with a few frames of labelled data as compared using the proposed SEAL framework for label propagation. We report bounding box and mask AP50 scores for Object Detection and Instance Segmentation.

| | Fine-tuning Mask-RCNN | | SEAL | |
|---|---|---|---|---|
| Num labels | Object Detection | Instance Segmentation | Object Detection | Instance Segmentation |
| 0 | 34.82 | 32.54 | 41.23 | 37.28 |
| 5 | 34.22 | 31.67 | 41.44 | 37.65 |
| 10 | 35.14 | 32.52 | 42.63 | 38.48 |

they are human-annotated. In Table 3, we report the performance of a Mask-RCNN naively fine-tuned with a few frames ($k = 0, 5, 10$) of labeled data as compared using the proposed SEAL framework for label propagation. Naively fine-tuning Mask-RCNN with 10 labeled examples does not improve the performance much. SEAL leads to a considerable performance improvement as labels are propagated to other observations using 3D semantic mapping.

## 6 Discussion

We presented the Self-supervised Embodied Active Learning (SEAL) framework for closing the action-perception loop. We demonstrated that the SEAL framework can be used to utilize a pretrained perception model to learn an active exploration policy for gathering useful observations. The observations gathered by the policy can be used to improve the pretrained perception model. Our ablation experiments highlight the importance of both the Action and Perception phases. The SEAL-improved perception can further be used to improve performance at Object Goal Navigation. The entire framework is completely self-supervised, the agent learns a policy and improves perception by just moving around in physical environments without having access to any human annotations.

The ability to learn in a self-supervised fashion comes at the cost of some limitations. The quality of the 3D semantic map (and the labels obtained from it) is dependent on the performance of the pretrained perception model. If the perception model never detects an object from any viewpoint, there is no way to learn about the object without any additional supervision. Similarly, if the perception model makes wrong predictions with high scores, these errors will be amplified by label propagation.

In the weak supervision extension, we explored how these limitations can be addressed to some extent with additional human supervision. In the future, the SEAL framework can also be extended to tackle few-shot learning of new objects with a few annotated examples. We studied this problem under the embodied active setting, but the self-supervised label propagation is also applicable to passive video data. We tackled static scenes in this paper. In the future, the 3D semantic map can potentially be extended to dynamic scenes with moving humans by explicitly predicting which voxels belong to static and dynamic objects.

## Acknowledgements

Carnegie Mellon University effort was supported in part by the US Army Grant W911NF1920104 and DSTA. UIUC effort is partially funded by NASA Grant 80NSSC21K1030. Jitendra Malik received funding support from ONR MURI (N00014-14-1-0671). Ruslan Salakhutdinov would also like to acknowledge NVIDIA's GPU support.

**Licenses for referenced datasets:**

Gibson: http://svl.stanford.edu/gibson2/assets/GDS_agreement.pdf

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
