# A   Pseudo Code

---

**Algorithm 1** Learning action

---

1: Initialize Dataset: $\mathcal{D} = \emptyset$
2: Initialize Pre-trained Peception Model: $f_{P;\theta}$
3: Initialize Gainful Curiosity Policy: $\pi_A; \omega$
4: E = Number of training environments
5: Initialize 3D Semantic Maps: $m_0 = \mathbf{0} \in \mathbb{R}^{E \times K \times L \times W \times H}$
6: T = Trajectory length
7: N = Number of training iterations
8: P = Number of RL epochs
9: **for** iteration $p = 1, 2, ...P$ **do**
10:    **for** iteration $e = 1, 2, ...E$ **do**
11:       $s_0^e = env.reset()$ {environment initial state}
12:       **for** iteration $t = 1, 2, ...T$ **do**
13:          $a_t = \pi_A(s_{t-1}^e)$
14:          $s_t^e = env.step(a_t)$ {environment step}
15:          $m_t^e = UpdateMap(m_{t-1}^e, s_t^e, f_{P;\theta})$
16:          $r^e = \text{sum}(m_t^e > 0.9)$
17:       **end for**
18:    **end for**
19: **end for**
20: **for** iteration $n = 1, 2, ...N$ **do**
21:    $\pi_A \leftarrow \nabla\mathbb{E}\left[\sum r\right]$
22: **end for**

---

**Algorithm 2** Learning perception

---

1: Initialize Dataset: $\mathcal{D} = \emptyset$
2: Initialize Pre-trained Peception Model: $f_{P;\theta}$
3: Initialize Trained Gainful Curiosity Policy: $\pi_A$
4: E = Number of training environments
5: T = Trajectory length
6: N = Number of training iterations
7: **for** iteration $e = 1, 2, ...E$ **do**
8:    Initialize 3D Semantic Map: $m_0 = \mathbf{0} \in \mathbb{R}^{K \times L \times W \times H}$
9:    $s_0^e = env.reset()$ {environment initial state}
10:    **for** iteration $t = 1, 2, ...T$ **do**
11:       $a_t = \pi_A(s_{t-1}^e)$
12:       $s_t^e = env.step(a_t)$ {environment step}
13:       $m_t = UpdateMap(m_{t-1}, s_t^e, f_{P;\theta})$
14:    **end for**
15:    $L^e = LabelMap(m_T)$ {Self-supervised labeling}
16:    **for** iteration $t = 1, 2, ...T$ **do**
17:       $I_t^e, D_t^e, x_t^e = s_t^e$ {RGB, Depth, Pose}
18:       $y_t^e = GetLabels(L^e, x_t^e, D_t^e)$ {RayTracing}
19:       $\mathcal{D} = \mathcal{D} \cup \{(I_t^e, y_t^e)\}$
20:    **end for**
21: **end for**
22: **for** iteration $j = 1, 2, ...N$ **do**
23:    sample batch $(I_k, y_k), ..., (I_{k+B}, y_{k+B})$
24:    update $\theta$ to minimize $\mathcal{L}(f_{P;\theta}(I_i), y_i)$ via SGD
25: **end for**

---

---

**Algorithm 3** Update Map

---

1: $I_t^e, D_t^e, x_t^e = s_t^e$ {RGB, Depth, Pose}
2: Compute agent centric point cloud (APC) from $D_t^e$ and $P$ camera matrix
3: Transform $x_t^e$ to geocentric pose $x_G^{e_t}$
4: Transform APC into geocentric point cloud (GPC) using $x_G^{e_t}$
5: Compute semantic obs $S_t^e$ as $f_{P;\theta}(I_t^e)$
6: Compute semantic features $f_t^e$: AveragePool ($S_t^e$)
7: Convert GPC into voxel grid and fill with $f_t^e$: $\hat{m}_t$
8: $m_t = \max(m_{t-1}, \hat{m}_t$

---

---

**Algorithm 4** Label Map

---

1: I = Number of total instances
2: $NCP$ = No category prediction threshold
3: Initialize $L^e \in \mathbb{R}^{I \times L \times W \times H}$
4: **for** iteration $k = 1, 2, ...K$ **do**
5:     thresh = $m_T[k] > NCP$
6:     thresh = RemoveSmallObjects(thresh)
7:     thresh = FillSmallHoles(thresh)
8:     thresh = BinaryDilate(thresh)
9:     $l$ = MorphologicalLabel(thresh)
10:     update $L^e$ with $l$
11: **end for**

---

---

**Algorithm 5** Get Labels

---

1: $H_V, W_V$ = height, width of voxel map
2: $H_I, W_I$ = height, width of desired ray traced image
3: $d_{min}, d_{max}$ = min, max depth to ray trace
4: Initialize $y_t^e$ to all zeros
5: Transform $m_t$ into agent centric map $m_t^a$ using $x_t^e$
6: **for** iteration $i = 0, ..., W_I$ **do**
7:     **for** iteration $k = 0, ..., H_I$ **do**
8:         Compute ray direction $r = atan(-(i - \frac{W_I}{2})/(\frac{W_I}{2})), atan(-(k - \frac{W_I}{2})/(\frac{W_I}{2}))$
9:         march along r and capture semantic map values to form image:
10:         **for** iteration $d = d_{min}, d_{min} + 1, ..., d_{max}$ **do**
11:             $p = [\frac{H_V}{2}, \frac{W_V}{2}] + d * \tan(r)$
12:             if $p$ inside voxel grid, $y_t^e[i, j] = m_t[p, d]$
13:         **end for**
14:     **end for**
15: **end for**

---

# B List of Training and Test scenes

| Dataset | Train split | | | | | Test split |
|---|---|---|---|---|---|---|
| Gibson | Allensville | Forkland | Leonardo | Newfields | Shelbyville | Collierville |
| | Beechwood | Hanson | Lindenwood | Onaga | Stockman | Corozal |
| | Benevolence | Hiteman | Marstons | Pinesdale | Tolstoy | Darden |
| | Coffeen | Klickitat | Merom | Pomaria | Wainscott | Markleeville |
| | Cosmos | Lakeville | Mifflinburg | Ranchester | Woodbine | Wiconisco |

# C Compute Requirements

We utilize 8 x 32GB V100 GPU system for training the active exploration policy using Gainful Curiosity and other Action baselines. We train the policy for 10 million frames, which takes around 2 days to train. The trajectories for the Perception phase are collected using single 32GB V100 GPU. It takes only a few minutes to collect each trajectory. The Mask-RCNN is fine-tuned using 8 x 32GB V100 GPUs. Fine-tuning the Mask-RCNN one takes less than 3 hrs. All the experiments are conducted on an internal cluster. The compute requirement can be reduced to single 16GB GPU by reducing the number of threads during policy training and reducing the batch size during Mask-RCNN training. Reducing compute will increase the training time.