# OpenReview forum: "SEAL: Self-supervised Embodied Active Learning using Exploration and 3D Consistency"
_NeurIPS.cc/2021/Conference — NeurIPS 2021 Poster_

### Official Review · Reviewer_RRT1 · 2021-07-13

**Rating:** 7
**Confidence:** 3

**Summary:**

The paper introduces “SEAL”, a framework for self-supervised learning of an embodied exploration policy while simultaneously refining the agent’s perception network. The method relies on building a semantic 3d map propagating confident predictions from one viewpoint to another for visual refinement. The method is validated on the Gibson dataset in terms of perceptual accuracy and on the downstream task of object goal navigation.


**Ethical Concerns:**

I could not identify any major ethical concerns.


**Limitations And Societal Impact:**

See my note regarding the limitations of the method since it relies on ground-truth pose and depth from the simulator.

**Main Review:**

Originality:

Paper builds on previous work and efficiently combines previously introduced concepts (intrinsic reward for exploration, self-supervised learning of object detector, training exploration policy using RL) into a holistic framework for self-supervised learning of embodied exploration and perceptual refinement.
This area of research is very active and I believe the authors have cited the most relevant work. The most similar work I could find was “Move to See Better: Towards Self-Supervised Amodal Object Detection” which is currently under review and cited by the authors. The methodology of the two papers differs significantly since this paper relies on building a 3d map of the environment and learning an exploration policy.

Quality:

In general the paper is technically sound and the idea interesting since self-supervised learning would enable life-long learning for robots in the field. The problem however with the paper is that it relies on ground-truth depth and position. An ablation on the effects of using predicted depth and a noisy position sensor would demonstrate the applicability of SEAL for real-world robotics beyond a simulation. If the current method requires ground-truth pose and depth then this limitation should be discussed.

Clarity:

The paper is generally clear and easy to read. However the result section could be improved upon to be easier to read. It took me some time to understand why you had two AP50 scores. Writing that one is for object detection and is for instance segmentation rather than implying it would improve the text.
I would also clarify early in the text what self-supervision means in your context. I.e. given a supervised pre-trained model (MaskRCNN) you can learn an exploration policy using self-supervision. The perceptual model itself is learned using a mix of supervised and self-supervised training.

Significance:

The significance of this paper lies in the novel idea of a self-supervised learning framework for embodied exploration and navigation. As a researcher I find this concept very appealing and the results on the downstream task verifies that it could have practical usefulness.


**Time Spent Reviewing:**

4

---

> ### Author Response · Authors · 2021-08-10
> **Response to Reviewer RRT1**
>
> We thank the reviewer for their valuable comments and suggestions. We address the concerns and answer your questions below:
>
> **Regarding Pose and Depth**: Prior work has shown that mapping-based navigation policies trained in simulation can be transferred successfully to mobile robot platforms in the real-world with noisy pose and depth sensors [9, 10]. We conducted more experiments with noisy pose and depth in the Habitat simulator. We use the actuation and motion pose noise models from [10] and the Redwood depth noise model from Choi et al. (CVPR 2015). The results are shown below:
>
> Table A2. **Noisy Pose and Depth Results.** Table showing results with noisy pose and depth on the Gibson dataset. We report bounding box and mask AP50 scores for Object Detection and Instance Segmentation.
>
> |                                   |   |  Generalization  |                       |   |  Specialization  |                       |
> |-----------------------------------|---|:----------------:|:---------------------:|:-:|:----------------:|:---------------------:|
> |                                   |   | Object Detection | Instance Segmentation |   | Object Detection | Instance Segmentation |
> | Best baseline numbers (Noiseless) |   |       35.85      |         32.57         |   |       35.90      |         33.29         |
> | SEAL (Noiseless)                  |   |       40.02      |         36.23         |   |       41.23      |         37.28         |
> | SEAL (Noisy Pose and Depth)       |   |       39.42      |         35.71         |   |       40.09      |         36.19         |
>
> The results indicate that even with noisy pose and depth, SEAL leads to much higher performance as compared to the best baseline.
>
> **Regarding clarity**: Thanks for the pointers to improve the clarity. We will add the relevant text describing two AP scores in the Results section and clarify what self-supervision means in our context.
>
> References:
>
> Choi, Sungjoon, Qian-Yi Zhou, and Vladlen Koltun. "Robust reconstruction of indoor scenes." CVPR 2015.
>
> [9] Devendra Singh Chaplot, Dhiraj Gandhi, Abhinav Gupta, and Ruslan Salakhutdinov. Object goal navigation using goal-oriented semantic exploration. In In Neural Information Processing Systems, 2020.
>
> [10] Devendra Singh Chaplot, Dhiraj Gandhi, Saurabh Gupta, Abhinav Gupta, and Ruslan Salakhutdinov. Learning to explore using active neural slam. In ICLR, 2020.

---

> > ### Author Response · Authors · 2021-08-23
> > **Question for Reviewer RRT1**
> >
> > We added additional experiments with noisy pose and depth to address the concern on reliance on ground-truth depth and position. However, we noticed that the rating was reduced from 7 to 6 and we could not find any other justification. Does our experiment not address your concern? We wanted to make sure this was not a misclick.

---

> > > ### Comment · Reviewer_RRT1 · 2021-08-26
> > > **Mistake**
> > >
> > > Sorry, I got confused with another review. I intended to keep my original score.

---

### Official Review · Reviewer_4ztR · 2021-07-16

**Rating:** 6
**Confidence:** 4

**Summary:**

This paper proposes an approach to fine-tune and transfer an object detection and segmentation approach (Mask-RCNN) to a set of 3D textured scene models by label propagation and exploration. The object detector is pretrained on an annotated large-scale Internet image dataset (MS Coco). A "perception" component maps the class pixel labels of Mask R-CNN into the 3D scene through raycasting in the given map and classifying voxels in a volumetric 3D semantic map. A policy is trained to explore the environment for fine-tuning the Mask-RCNN detector.

**Ethical Concerns:**

none apparent.

**Limitations And Societal Impact:**

Several limitations of the work are well discussed in Sec. 6. See paper weaknesses above for unaddressed shortcomings (instance segmentation not used).



**Main Review:**

Strengths:
- The paper proposes an interesting novel combination of 3D semantic mapping, label propagation and exploration using a goal policy network.
- The experiments demonstrate the approach on a benchmark dataset and show good results, improving over several state of the art baselines and ablations.

Weaknesses:
- A voxel-based semantic representation is accumulated by classifying the voxels according to the highest classification score of the pixels striding the voxel. It seems the instance segmentation of Mask R-CNN is not used and individual objects cannot be distinguished. This would make the method fail in scenes where multiple instances of a class (like chairs) are close and would be detected as one instance, since Mask R-CNN would receive wrong instance segmentation labels for training.
- The method assumes noiseless depth and pose information which is unrealistic. How does the method perform if real depth images and a localization estimator need to be used ?
- Which optical flow method is used as a baseline? Please compare against a state-of-the-art method like PWC-Net.
- Please specify the memory and runtime requirements at test time for the Action model.
- It's not clear what the "Random Policy" is: does the agent execute a random move action, or navigate to random goal locations and take images there ? I think the latter would be clearly superior.
- When does each baseline method stop collecting images? Is the same number of images collected per baseline? Are the comparisons between methods fair ?

== Post-rebuttal rating ==

The author response well addressed most of my concerns. The paper contains limited but sufficient novelty over previous methods which perform active SLAM or simultaneously train the object detector during SLAM  by presenting an actor component which is trained for semantic exploration. I increase my score to borderline 6 "marginally above acceptance threshold".

**Time Spent Reviewing:**

4

---

> ### Author Response · Authors · 2021-08-10
> **Response to Reviewer 4ztR**
>
> We thank the reviewer for their valuable comments and suggestions. We address the concerns and answer the questions below:
>
> - *”A voxel-based semantic representation is accumulated by classifying the voxels according to the highest classification score of the pixels striding the voxel. It seems the instance segmentation of Mask R-CNN is not used and individual objects cannot be distinguished. This would make the method fail in scenes where multiple instances of a class (like chairs) are close and would be detected as one instance, since Mask R-CNN would receive wrong instance segmentation labels for training.”*
>
> This is a great observation. If multiple instances of a class are connected to each other, they would be labelled as a single instance by our 3DLabelProp. However, in our experience this happens very rarely (only for dining chairs in some scenes). This is evident from the strong performance numbers. Despite this limitation, SEAL improves performance of the pretrained Mask-RCNN from 34.82/32.54 AP50 (for object detection and instance segmentation) to 40.02/36.23 without requiring any label or any experience in the test environment in a completely self-supervised fashion. In future, this limitation can be addressed by tracking the Mask RCNN predictions over the exploration trajectory, to build the semantic map based on instance predictions and not just semantic categories.
>
> - *”The method assumes noiseless depth and pose information which is unrealistic. How does the method perform if real depth images and a localization estimator need to be used ?”*
>
> Prior work has shown that mapping-based navigation policies trained in simulation can be transferred successfully to mobile robot platforms in the real-world with noisy pose and depth sensors [9, 10]. We conducted more experiments with noisy pose and depth in the Habitat simulator. We use the actuation and motion pose noise models from [10] and the Redwood depth noise model from Choi et al. (CVPR 2015). The results are shown below:
>
> Table A2. **Noisy Pose and Depth Results.** Table showing results with noisy pose and depth on the Gibson dataset. We report bounding box and mask AP50 scores for Object Detection and Instance Segmentation.
>
> |                                   |   |  Generalization  |                       |   |  Specialization  |                       |
> |-----------------------------------|---|:----------------:|:---------------------:|:-:|:----------------:|:---------------------:|
> |                                   |   | Object Detection | Instance Segmentation |   | Object Detection | Instance Segmentation |
> | Best baseline numbers (Noiseless) |   |       35.85      |         32.57         |   |       35.90      |         33.29         |
> | SEAL (Noiseless)                  |   |       40.02      |         36.23         |   |       41.23      |         37.28         |
> | SEAL (Noisy Pose and Depth)       |   |       39.42      |         35.71         |   |       40.09      |         36.19         |
>
> The results indicate that even with noisy pose and depth, SEAL leads to a much higher performance as compared to the best baseline.
>
> - *”Which optical flow method is used as a baseline? Please compare against a state-of-the-art method like PWC-Net.”*
>
> We use the ground-truth optical flow from the depth and pose which we believe is superior to any learning-based method for optical flow.
>
> - *”Please specify the memory and runtime requirements at test time for the Action model.”*
>
> We run the Action model on a single GPU of 16GB memory which results in 5 frames per second. Running the model on CPU results in around 3 frames per second. This is faster than mobile robot platforms such as the Locobot which operates at less than 0.5 actions per second in the real-world.
>
> - *”It's not clear what the "Random Policy" is: does the agent execute a random move action, or navigate to random goal locations and take images there ? I think the latter would be clearly superior.”*
>
> Thanks for the suggestion. The random policy is just picking random actions. We implemented a Global Random Policy + Optical Flow baseline and it achieves 33.98/32.85 AP50 for Generalization, 33.95/32.24 AP50 for Specialization, as compared to 40.02/36.23 (Generalization) and 41.23/37.28 (Specialization) for SEAL.
>
> - *”When does each baseline method stop collecting images? Is the same number of images collected per baseline? Are the comparisons between methods fair ?”*
>
> For the Action phase, the SEAL Gainful Curiosity policy and the Active Neural SLAM and Semantic Curiosity baselines are all trained for 10 million frames each. Other Action baselines are based on heuristics and do not require any training. For the perception phase, all baselines and SEAL collect a single episode of images (T=300) per environment. All methods collect exactly the same number of images.
>
> References:
>
> Choi, Sungjoon, Qian-Yi Zhou, and Vladlen Koltun. "Robust reconstruction of indoor scenes." CVPR 2015.
>
> [9] Devendra Singh Chaplot, Dhiraj Gandhi, Abhinav Gupta, and Ruslan Salakhutdinov. Object goal navigation using goal-oriented semantic exploration. In In Neural Information Processing Systems, 2020.
>
> [10] Devendra Singh Chaplot, Dhiraj Gandhi, Saurabh Gupta, Abhinav Gupta, and Ruslan Salakhutdinov. Learning to explore using active neural slam. In ICLR, 2020.

---

> > ### Author Response · Authors · 2021-08-23
> > **Note to Reviewer 4ztR**
> >
> > In our response, we provided additional ablations on noisy pose and depth, additional baselines and details requested about training and evaluation. Please let us know if our response does not address your concerns.

---

### Official Review · Reviewer_HudJ · 2021-07-17

**Rating:** 5
**Confidence:** 4

**Summary:**

The proposed method aims to improve the performance of a pretrained perception model in an active learning way. The whole training framework consists of the Action phase and Perception phase, which are trained with gainful curiosity reward and pseudo labels projected from that of the 3D Map, separately. The experiments conducted on detection, segmentation, and object goal navigation tasks illuminate that they can achieve a stronger perception which benefits the embodied robotic task (Object Goal Navigation) in static scenes.

**Limitations And Societal Impact:**

The weaknesses of proposed methods are discussed and addressed by weak supervision.

**Main Review:**

Strength：
- The motivation is intuitive and reasonable as well as and the results look great.
- It is well written and easy to understand the design of gainful curiosity reward and the way to get pseudo labels.
- The weaknesses of proposed methods are discussed and addressed by weak supervision.

Weakness:
-  It is not a new concept to employ spatial-temporal consistency to generate pseudo labels for fine-tuning the perception module. It would be better to discuss the difference to related work, e.g., [1], [2].
-  If my understanding serves me right, the framework is carried out by two steps sequentially, instead of an interactive learning style, which seems more reasonable for me.
-  Missing some details, like the evaluation metric
-  Some typos or mistakes, like ‘to used to’ in Line160.

Detail comments:
-  It is not clear whether this method is applicable to all embodied robots. If it can, why not evaluate it on more tasks. If not, can it perform well on some common dataset for the segmentation task? Or does it have its own significant advantages for Object Goal Navigation? If there are some moving objects in the scene, can the method still work well?
-  Although the proposed method outperforms the baseline methods on some metrics, like AP50, according to the video, it seems that the proposed algorithm tends to cover a larger area (even some wrong areas) compared with Mask RCNN while predicting segmentation. Can you give an explanation?
-  The Gainful reward designing seems intuitive, do you use the number of voxels in the 3D Map (value> score threshold) directly, without any processing?



References:

[1] Zhong, Fangwei, et al. "Detect-slam: Making object detection and slam mutually beneficial." WACV 2018.

[2] Wang, Kai, et al. "A unified framework for mutual improvement of slam and semantic segmentation." ICRA 2019.



**Time Spent Reviewing:**

8

---

> ### Author Response · Authors · 2021-08-10
> **Response to Reviewer HudJ**
>
> We thank the reviewer for their valuable comments and suggestions. We address the concerns and answer the questions below:
>
> - *”It is not a new concept to employ spatial-temporal consistency to generate pseudo labels for fine-tuning the perception module. It would be better to discuss the difference to related work, e.g., [1], [2].”*
>
> Thanks for pointing out the relevant prior work. [1] and [2] both fall in the category of active perception methods, i.e., they learn active interaction with an environment to improve performance at test time by gaining information from better views. What separates our work from [1] and [2] is that our work focuses on active learning through interaction, i.e. it learns how to explore an environment at train time to automatically generate more training data which is used to improve perception. We will add [1] and [2] to the related work section in our discussion on “Active Perception and Learning” (L82-104).
>
> - *”If my understanding serves me right, the framework is carried out by two steps sequentially, instead of an interactive learning style, which seems more reasonable for me.”*
>
> We are not sure what you mean by an interactive learning style, could you elaborate? Perhaps, the difference is active perception vs active learning as described above? We believe both problems are different and important. Active perception allows better test time adaption (in perhaps a more interactive style). Active learning tackles learning in training environments allowing continuous improvement perception across multiple environments and can achieve better generalization performance in test environments.
>
> - *”Missing some details, like the evaluation metric”*
>
> We use bounding box and mask AP50 scores as the evaluation metric for object detection and instance segmentation, respectively, as mentioned in Section 4 (L227-228). AP50 is the average precision with at least 50% IOU. IOU is defined to be the intersection over union of the predicted and ground-truth bounding box or the segmentation mask. We will add the description of the evaluation metric in Section 4.
>
> - *”It is not clear whether this method is applicable to all embodied robots. If it can, why not evaluate it on more tasks. If not, can it perform well on some common dataset for the segmentation task? Or does it have its own significant advantages for Object Goal Navigation? If there are some moving objects in the scene, can the method still work well?”*
>
> We believe the method is applicable to any embodied robot with simple RGBD + pose sensors. We would like to highlight that we evaluate our method on many more tasks than prior work. We evaluate the method for two perception tasks, i.e. object detection and instance segmentation and for a challenging action task, i.e. object goal navigation. We also evaluate the method for few-shot learning in the weak supervision setting. In comparison, prior published work on this problem [11] only evaluates the method on the object detection task.
> Regarding dynamic objects, as stated in the Discussion section, the 3D semantic map can potentially be extended to dynamic scenes by explicitly predicting which voxels belong to static and dynamic objects (L347-349). Tackling dynamic objects is beyond the scope of the current work as the Habitat simulator does not have the ability to simulate dynamic objects.
>
> - *”Although the proposed method outperforms the baseline methods on some metrics, like AP50, according to the video, it seems that the proposed algorithm tends to cover a larger area (even some wrong areas) compared with Mask RCNN while predicting segmentation. Can you give an explanation?”*
>
> This is a limitation that stems from self-supervised learning. As stated in the Discussion section, the quality of the 3D semantic map (and the labels obtained from it) is dependent on the performance of the pretrained perception model. If the perception model never detects an object from any viewpoint, there is no way to learn about the object without any additional supervision. Similarly, if the perception model makes wrong predictions with high scores, these errors will be amplified by label propagation. In some video examples, the method covers a larger area as the pretrained MaskRCNN is making some wrong mask prediction with high scores from some viewpoints.
> In spite of this limitation, our method achieves a much higher performance than a pretrained MaskRCNN, indicating that the pretrained MaskRCNN makes more correct predictions with high scores than wrong predictions and on average, label propagation helps significantly. Furthermore, in the weak supervision section, we explored how these limitations can be addressed using additional human supervision.
>
> - *”The Gainful reward designing seems intuitive, do you use the number of voxels in the 3D Map (value> score threshold) directly, without any processing?”*
>
> We multiply the number of voxels in the 3D map by a coefficient of 0.001 to compute the reward.
>
> References:
>
> [1] Zhong, Fangwei, et al. "Detect-slam: Making object detection and slam mutually beneficial." WACV 2018.
>
> [2] Wang, Kai, et al. "A unified framework for mutual improvement of slam and semantic segmentation." ICRA 2019.
>
> [11] Devendra Singh Chaplot, Helen Jiang, Saurabh Gupta, and Abhinav Gupta. Semantic curiosity for active visual learning. In ECCV, 2020.

---

> > ### Author Response · Authors · 2021-08-23
> > **Note to Reviewer HudJ**
> >
> > In our response, we provided the details requested about the method and evaluation and answered the questions. Please let us know if any of your concerns were not addressed by our response.

---

### Official Review · Reviewer_9kyX · 2021-07-19

**Rating:** 6
**Confidence:** 4

**Summary:**


This paper presents a technique for improving a pre-trained object detection model by learning an exploration policy in a 3D environment, automatically labeling the experience collected by this policy, and then using this data to finetune the object detection model. This improves the object detection performance of the pretrained model, and using this finetuned model in an agent leads to higher success in object navigation tasks.

The exploration policy operates on a 3D voxel-based map representation. Each voxel contains one number per object category (e.g., chair, bed) denoting the confidence that an object of that type is present in that voxel. This representation is built by using a pre-trained MaskRCNN and using depth information to map back pixels in 2D space to 3D voxels. This voxel-level object confidence scores are summed over time to create the 3D (voxel-based) semantic map of the environment. To learn the exploration policy, they use a curiosity signal named Gainful Curiosity. This essentially counts the number of voxels with a confident object prediction. The policy consists of 2 levels with the high level Global policy proposing a new goal every 25 timesteps, and a low level Local policy that uses a path finding algorithm to navigate to the proposed goal. The global policy is trained to maximize the reward obtained from the Gainful Curiosity signal.

The experience collected by the exploration policy is then used to finetune the object detection model. To do this, they introduce a label propagation technique called 3DLabelProp. This essentially looks at a set of connected voxels with the same label and constructs the 2D bounding boxes/masks by mapping back from 3D map to 2D observations. Then they train the object detection model on observations labeled in this way.

They evaluate their technique on a simulated house environment and show that they can improve object detection/segmentation performance of a pre-trained MaskRCNN. They also show that this improved object detection model can then be used to achieve higher success in object goal navigation.


**Ethical Concerns:**

No ethical concerns.

**Limitations And Societal Impact:**

Yes.

**Main Review:**


The presented technique is relatively straightforward and perhaps not very novel. However, the motivation is I think quite solid; it is obvious that the 3D spatio-temporal structure of the world provides valuable information to our perceptual systems and this works shows one simple way how this can be exploited to improve object detection/segmentation. I think overall the paper is technically sound and the evaluations look OK. It is also clearly written and easy to follow.

I think the main issue with the paper is that the presented technique itself is quite simple. This is in itself not necessarily a problem of course. However, if the presented technique is simple, I'd expect much more evidence that even the technique itself is very simple, it has very wide applicability and results in quite significant improvements on a wide variety of environments. As it stands, I think the paper unfortunately fails to show this.

Please see my other comments below:
- Does the local policy (fast marching method) know about the 3D layout of the house (e.g., walls, doors etc.)?
- In the ablation study, what is random policy? Is this just picking random actions? Or picking goals randomly (positions from 3D map)? If it is the former, I think doing the latter (sample goals randomly and navigate to them) would be a better test of how much the gainful curiosity helps.
- For the object navigation experiment that shows you get higher success with finetuned MaskRCNN, isn't this pretty much expected? I feel like a better comparison could be to train a policy to navigate to objects from RGB-D data (or the 3D semantic map with a standard/pre-trained MaskRCNN). Since your technique uses RGB-D data to improve the perceptual model, it seems fairer to provide the same info to the baseline as well.
- typo, line 133, "both the phases" -> "both phases" or "both of the phases"

**Time Spent Reviewing:**

3

---

> ### Author Response · Authors · 2021-08-10
> **Response to Reviewer 9kyX**
>
> We thank the reviewer for their valuable comments and suggestions. We address the concerns and answer the questions below:
>
> **Regarding simplicity**: We would like to argue that the simplicity of a method is an advantage and not a disadvantage. It makes our method easy to understand and implement. We would like to highlight that simplicity of the method does not imply that it is not novel. We are not aware of any other work which proposes such a simple method for this important task, which the reviewer agrees is well-motivated. More importantly, our results show that our simple method achieves exceptional results improving pretrained Mask-RCNN performance from 34.82/32.54 AP50 (for object detection and instance segmentation) to 40.02/36.23 without requiring any label or any experience in the test environment in a completely self-supervised fashion. The best baseline can only achieve 35.85/32.57.
>
> **More environments**: We provide results training with 25 environments and testing in 5 different environments in the Gibson dataset in the paper. We would like to emphasize that the environments used in our experiments are very diverse and challenging as shown in the video demonstrations in the supplementary material. We conducted additional experiments with the Replica dataset. For the following results, we train all the methods in 25 Gibson training environments + 9 environments in Replica and test on 4 unseen environments in Replica (we exclude 5 environments which are different rearrangements of the same objects in the frl_apartment environment).
>
> Table A1. Table showing results on the Replica dataset. We report bounding box and mask AP50 scores for Object Detection and Instance Segmentation.
>
> |                                    |   |  Generalization  |                       |   |  Specialization  |                       |
> |-----------------------------------|---|:----------------:|:---------------------:|---|:----------------:|:---------------------:|
> | Method                             |   | Object Detection | Instance Segmentation |   | Object Detection | Instance Segmentation |
> | Pretrained Mask-RCNN               |   |       45.98      |         38.76         |   |       45.98      |         38.76         |
> | Random Policy + Self-training      |   |       45.29      |         38.15         |   |       45.62      |         38.29         |
> | Random Policy + Optical Flow       |   |       45.44      |         38.29         |   |       45.9       |         38.87         |
> | Active Neural SLAM + Self-training |   |       45.99      |         37.89         |   |       46.87      |         38.90         |
> | Active Neural SLAM + Optical Flow  |   |       46.28      |         38.92         |   |       47.19      |         39.12         |
> | Semantic Curiosity + Self-training |   |       46.90      |         39.10         |   |       47.21      |         39.62         |
> | Semantic Curiosity + Optical Flow  |   |       46.81      |         39.19         |   |       46.89      |         39.52         |
> | SEAL                               |   |       **51.29**      |         **43.95**         |   |       **52.30**      |         **46.26**         |
>
> As shown in the results above, SEAL outperforms all the baselines in the Replica environments as well, achieving 51.29/43.95 AP50 scores as compared to 46.90/39.19 for the best baseline and 45.98/38.76 for pretrained Mask-RCNN. Allowing a single episode of exploration in the test environments (in the Specialization setting) further improves the performance to 52.30/46.26 AP50 scores for SEAL as compared to 47.21/39.62 for the best baseline.
>
> **Other questions**:
> - *"Does the local policy (fast marching method) know about the 3D layout of the house (e.g., walls, doors etc.)?"*
>
> The local policy uses top-down 2D maps, it consists of obstacles like walls but not in 3D. The top-down 2D map is built by summing over the height channel in the 3D map. Note that the map is not provided to the local policy and is built on the fly using depth observations.
>
> - *"In the ablation study, what is random policy? Is this just picking random actions? Or picking goals randomly (positions from 3D map)? If it is the former, I think doing the latter (sample goals randomly and navigate to them) would be a better test of how much the gainful curiosity helps."*
>
> Thanks for the suggestion. The random policy is just picking random actions. We implemented a SEAL w/o Action + Random Global Policy ablation as well and it achieves 35.98/32.55 AP50 for Generalization, 36.19/32.87 AP50 for Specialization, as compared to 40.02/36.23 (Generalization) and 41.23/37.28 (Specialization) for SEAL.
>
> - *"For the object navigation experiment that shows you get higher success with finetuned MaskRCNN, isn't this pretty much expected? I feel like a better comparison could be to train a policy to navigate to objects from RGB-D data (or the 3D semantic map with a standard/pre-trained MaskRCNN). Since your technique uses RGB-D data to improve the perceptual model, it seems fairer to provide the same info to the baseline as well."*
>
> The SemExp baseline [9] also utilizes RGB-D data with a standard/pre-trained MaskRCNN for building a semantic map and learning the semantic exploration policy. Our sensor setup is identical to the SemExp baseline.
>
>
> References:
>
> [9] Devendra Singh Chaplot, Dhiraj Gandhi, Abhinav Gupta, and Ruslan Salakhutdinov. Object goal navigation using goal-oriented semantic exploration. In In Neural Information Processing Systems, 2020

---

> > ### Comment · Reviewer_9kyX · 2021-08-23
> > **Thanks for the response**
> >
> > I'd like to thank the authors for their detailed response. After reading their response and other reviews, I think the authors have addressed the main points and I see there is enough novelty for the work to be interesting to the wider community. I will update my score accordingly.

---

### Decision · Program_Chairs · 2021-09-27

**Decision:**

Accept (Poster)

**Comment:**

This paper proposes a method which, like active vision, combines perception and control in a closed loop to improve perception (and also an agent policy). It leverages data from pre-trained models, which are bootstrapped, and proposes two different settings, generalization vs. specialization, which also allows training on the test environment.

The paper received 4 expert reviews and was initially on the fence. The reviewers initially mainly raised issues on lack of novelty, an requested additional experiments.

The authors provided a response, which was highly appreciated by most reviewers and solved several issues, in particular through a fruitful discussion on novelty. Additional experiments also help the paper. Some reviewers raised the score above borderline, and 3 out of the 4 reviewers were convinced on acceptance.

The AC followed the discussion, and while he does not find the paper particularly well-written, concurs that it has merits and recommends acceptance.